# Structural basis of respiratory syncytial virus subtype-dependent neutralization by an antibody targeting the fusion glycoprotein

Daiyin Tian[1,2], Michael B. Battles [3], Syed M. Moin[1], Man Chen[1], Kayvon Modjarrad[4,5], Azad Kumar[1], Masaru Kanekiyo [1], Kevin W. Graepel[6], Noor M. Taher[3], Anne L. Hotard[7,10], Martin L. Moore[7,11], Min Zhao[8,9], Zi-Zheng Zheng[8,9], Ning-Shao Xia[8,9], Jason S. McLellan[3] & Barney S. Graham [1]

A licensed vaccine for respiratory syncytial virus (RSV) is unavailable, and passive prophylaxis with the antibody palivizumab is restricted to high-risk infants. Recently isolated antibodies 5C4 and D25 are substantially more potent than palivizumab, and a derivative of D25 is in clinical trials. Here we show that unlike D25, 5C4 preferentially neutralizes subtype A viruses. The crystal structure of 5C4 bound to the RSV fusion (F) protein reveals that the overall binding mode of 5C4 is similar to that of D25, but their angles of approach are substantially different. Mutagenesis and virological studies demonstrate that RSV F residue 201 is largely responsible for the subtype specificity of 5C4. These results improve our understanding of subtype-specific immunity and the neutralization breadth requirements of next-generation antibodies, and thereby contribute to the design of broadly protective RSV vaccines.

[1] Vaccine Research Center, National Institute of Allergy and Infectious Diseases, National Institutes of Health, Bethesda, MD 20892, USA. [2] Department of Pulmonology, Children's Hospital of Chongqing Medical University, Chongqing 400014, China. [3] Department of Biochemistry and Cell Biology, Geisel School of Medicine at Dartmouth, Hanover, NH 03755, USA. [4] Walter Reed Army Institute of Research, Silver Spring, MD 20910, USA. [5] Henry M. Jackson Foundation, Bethesda, MD 20817, USA. [6] Department of Pediatrics, Vanderbilt University Medical Center, Nashville, TN 37208, USA. [7] Department of Pediatrics, Emory University, Atlanta, GA 30322, USA. [8] State Key Laboratory of Molecular Vaccinology and Molecular Diagnostics, School of Public Health, Xiamen University, Xiamen, Fujian 361005, China. [9] National Institute of Diagnostics and Vaccine Development in Infectious Diseases, School of Life Sciences, Xiamen University, Xiamen, Fujian 361005, China. [10] Present address: Division of Select Agents and Toxins, Centers for Disease Control and Prevention, Atlanta, GA 30333, USA. [11] Present address: Meissa Vaccines, Inc South, San Francisco, CA 94080, USA. Daiyin Tian and Michael B. Battles contributed equally to this work. Correspondence and requests for materials should be addressed to N.-S.X. (email: nsxia@xmu.edu.cn) or to J.S.M. (email: Jason.S.McLellan@Dartmouth.edu) or to B.S.G. (email: bgraham@nih.gov)

Respiratory syncytial virus (RSV) is an enveloped, negative-sense RNA virus that is the most important lower respiratory tract pathogen of children below 5 years of age[1] and is second only to malaria as a cause of death by a single pathogen in children <1 year of age[2]. Although the virus infects nearly all children by the age of three[3] and causes repeated infections throughout life[4], an effective vaccine is unavailable. The failure of natural infection to provide durable immunity is not explained by the genetic diversity of RSV, which has two major antigenic subtypes, A and B[5]. These subtypes co-circulate annually with relatively equal frequencies[6], but there is considerable debate as to whether clinical severity is impacted by the subtype of the infecting RSV strain[7–11]. RSV subtypes vary primarily in the mucin-like domains of the attachment (G) glycoprotein[12], but the fusion (F) glycoprotein is the primary target for neutralizing antibodies.

RSV F is highly conserved between subtypes with only ~ 30 amino acid differences in the mature ectodomain among subtype consensus sequences. RSV F is a class I fusion glycoprotein that is synthesized as an inactive precursor (F0) that is processed by furin-like proteases at two sites to generate three polypeptides: the N-terminal fragment (F2), a 27-amino-acid glycopeptide (pep27) and the C-terminal fragment (F1). F1 contains all the elements needed to promote fusion, including the fusion peptide (FP), two heptad repeats, and the transmembrane domain. The mature, active protein exists as a trimer of F2–F1 heterodimers, folded into a compact prefusion conformation (pre-F) on the viral envelope[13]. Pre-F undergoes a conformational transition to the elongated and highly stable postfusion conformation (post-F). During this refolding event, the hydrophobic FP is inserted into the host-cell membrane, and the viral and host membranes are fused, allowing delivery of the RSV genome into the cell.

RSV is sensitive to neutralization by antibodies directed against F. Neutralizing antibodies target six known antigenic sites: two of which (sites Ø and V) are considered pre-F-specific and four of which (sites I, II, III and IV) are available to various extents on both pre-F and post-F[14]. The only licensed product available for RSV prophylaxis is palivizumab (Synagis®), which is an RSV F-specific monoclonal antibody (mAb) that recognizes antigenic site II and is equally effective against RSV strains from subtype A and B[15]. Recently, mAbs recognizing the pre-F-specific antigenic site Ø have been discovered that have much greater neutralization potency than palivizumab[13, 16]. Since at least six of the subtype-specific substitutions in the F ectodomain fall within antigenic site Ø, it is critical to understand the neutralization breadth for mAbs targeting this site[13]. Thus, two potently neutralizing site Ø-specific mAbs, 5C4, and D25, are compared in the present study. Site Ø-specific mAbs are important because of their potential value for passive prophylaxis, and because they identify this antigenic site as a key site of RSV vulnerability that will be a crucial antigenic component of future vaccines. 5C4 was elicited in mice immunized with DNA and recombinant adenovirus expressing RSV F, and was identified by screening hybridomas for neutralizing activity (positive selection) and binding to recombinant post-F (negative selection), where each of these steps utilized F protein sequences derived exclusively from subtype A virus[13, 17]. Conversely, D25 was isolated from human B cells derived from an adult donor likely infected throughout life with RSV strains of both A and B subtypes[18]. Observed differences in the neutralization breadth of these mAbs present an opportunity to investigate the structural basis for subtype-dependent recognition of this important antigenic site.

Here we present the structure of 5C4 in complex with RSV F and explore the determinants of 5C4 binding. We show that 5C4 potently neutralizes a panel of RSV subtype A strains, yet poorly neutralizes subtype B strains, whereas D25 potently neutralizes

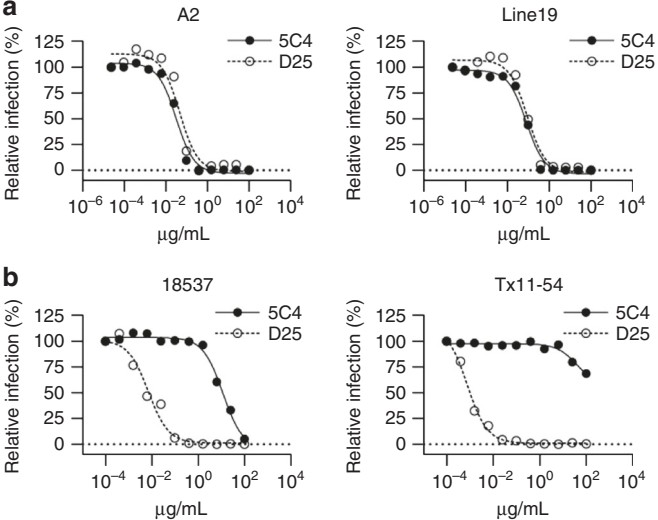

**Fig. 1** 5C4 is a potent subtype A-specific neutralizing antibody. **a** Neutralization potency of 5C4 (closed circles) and D25 (open circles) against RSV subtype A strains A2 (left) and Line19 (right). **b** Neutralization potency of 5C4 and D25 against subtype B strains 18537 (left) and Tx11-54 (right), depicted as in **a**. The data in panels **a** and **b** are representative of two independent experiments performed in duplicate

strains of both RSV subtypes. Our structural, binding, and in vitro neutralization analyses reveal that the subtype-dependent neutralization profile of 5C4 is likely due to a requirement for a positively charged residue at RSV F position 201. Our results demonstrate that slight antigenic variation within site Ø can cause mAbs to have restricted neutralization profiles and suggest that care should be taken to avoid biasing the antigenic repertoire towards a specific RSV strain or subtype during vaccination or therapeutic mAb selection.

## Results

**The neutralization potency of 5C4 is subtype-dependent**. 5C4 was previously shown to potently neutralize RSV strain A2, with a potency similar to other pre-F-specific antibodies such as D25 and AM22[13]. However, when analyzed against a panel of RSV strains including both subtypes A and B, it was revealed that 5C4 had significantly reduced neutralization potency against various subtype B viruses (Fig. 1a, b), as well as reduced affinity towards pre-F derived from subtype B strain 9320, as measured by surface plasmon resonance (Supplementary Fig. 1). As antibodies against pre-F provide the dominant neutralizing response against RSV[19], we sought to inform the design of future vaccines by determining the mechanisms of subtype-specific neutralization for this potent pre-F-specific antibody.

**A conserved mode of binding of 5C4 and D25 to RSV pre-F**. To define the 5C4 epitope on RSV F, the crystal structures of 5C4 alone and in complex with RSV F were determined. Crystals of the 5C4 Fab diffracted X-rays to 1.5 Å resolution. A molecular replacement solution was obtained and the structure was built and refined to an $R_{work}/R_{free}$ of 17.0%/20.3% (Supplementary Table 1, Supplementary Fig. 2a). In parallel, a construct consisting of the wild-type A2 RSV F ectodomain fused to a C-terminal trimerization motif was co-expressed with the 5C4 Fab and purified by affinity and gel-filtration chromatography. To obtain a structure of the 5C4 Fab bound to RSV F, an initial crystallization hit was optimized with additives and microseed matrix screening to yield a single crystal that diffracted X-rays to 3.4 Å resolution.

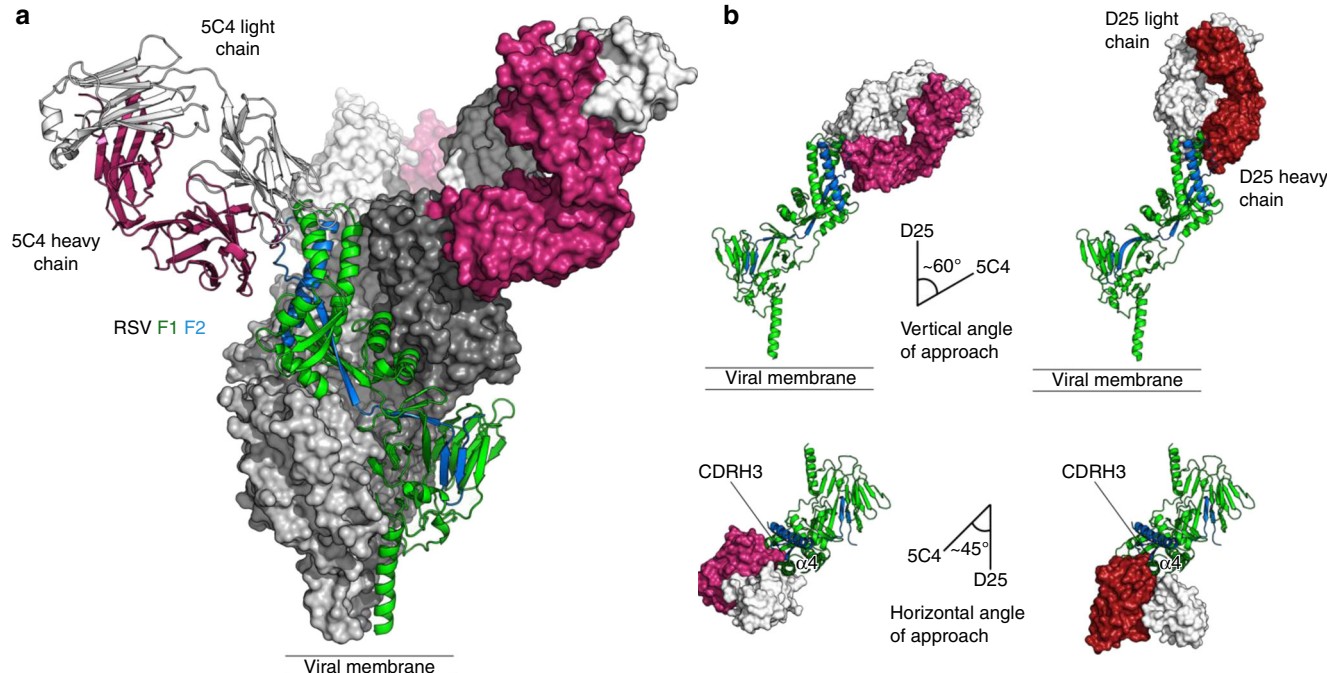

**Fig. 2** Mode of 5C4 binding to antigenic site Ø is similar to that of D25. **a** 5C4–RSV F complex crystal structure in ribbon and molecular surface representation. Two pre-F protomers are shown as dark and light gray surfaces, and the third protomer is shown as ribbons with F1 colored green and F2 colored blue. The 5C4 Fab bound to the F protomer is also displayed in ribbon representation, with the heavy chain colored pink and the light chain colored white. The other 5C4 Fabs are colored similarly but shown in surface representation. **b** RSV F protomers are shown in ribbon representation and colored as above. Fabs (upper) and antibody variable domains (lower) are shown as a molecular surface with light chains colored white and the heavy chains of 5C4 and D25 colored pink and red, respectively. The difference in vertical (upper) and horizontal (lower) angle of approach between 5C4 and D25 is represented as intersecting lines with a measured angle of ~ 60° and ~ 45°, respectively

The complex of 5C4–RSV F was solved by molecular replacement using the unbound 5C4 Fab structure and an unbound pre-F A2 structure (PDB ID 5C69[20]) as search models. The structure was refined to an $R_{work}/R_{free}$ of 19.8%/23.9%, and the composition of the asymmetric unit was identical to the biological unit and consisted of one trimer of RSV F bound to three 5C4 Fabs (Fig. 2a and Supplementary Table 1). The observed stoichiometry was consistent with the apparent molecular weight observed by size-exclusion chromatography (Supplementary Fig. 3) and previously obtained electron micrographs[13]. Although the F glycoprotein was fully glycosylated, the electron density at each of the N-linked glycosylation sites was insufficient to accurately model the glycans (Supplementary Fig. 2b).

The 5C4–RSV F interface is situated at the apex of the pre-F trimer at the previously identified antigenic site Ø[13]. Each of the three 5C4 Fabs buries an average of ~ 1000 Å$^2$ of solvent-accessible surface area on RSV F, with the heavy chain burying approximately two-thirds of this area. In addition, the shape-complementarity score for 5C4 and RSV F is 0.64 (a value of 1 representing a perfect fit), which is in the range of most antibody–antigen interactions and is nearly identical to the shape-complementarity score of 0.63 for D25 and RSV F[21]. The overall binding mode of 5C4 is similar to that observed for D25 (PDB ID 4JHW[13]) in that the α4 helix of F1 is clasped between the heavy chain third complementarity-determining region (CDRH3) and the light chain CDRs, although the vertical and horizontal angles of approach differ by 60° and 45°, respectively (Fig. 2b).

The antibody combining site of 5C4 is balanced in that it uses sparsely distributed contacts to engage its epitope, forming hydrogen bonds to RSV F with five of the six CDR loops (Fig. 3a). The 5C4 CDRH1 and CDRH2 loops each form a single hydrogen bond with the RSV F2 subunit, contacting the side chains of

Lys65 and Asn63, respectively. Asp52 in the CDRH2 also forms a salt bridge with the side chain of Lys65 in F2. The CDRH3 is wedged between the α4 helix and the β2–α1 loop, and main-chain atoms in the CDRH3 form hydrogen bonds to RSV F residues Asn208 and Lys68. The CDRH3 buries a high fraction of the accessible surface area on the α4 helix, including the majority of the accessible surface area on the Asp200 and Asn208 side chains (Supplementary Fig. 4). The 5C4 CDRL1 is of normal length for a murine Vk3-10 sequence and yet contributes 6 of the 13 observed hydrogen bonds at the 5C4–RSV F interface. These include hydrogen bonds to the Lys201 and Gln202 main-chain carbonyls, and the Gln202 and Lys209 side chains. The Asp29 residue forms two additional hydrogen bonds using both main- and side-chain atoms to contact both the carbonyl and amino groups of the RSV F Gln202 side chain. The 5C4 CDRL3 also forms a single hydrogen bond with the side chain of Lys201 in F1. D25, which neutralizes A and B subtypes equally well, forms hydrogen bonds with more than twice as many main-chain atoms in RSV F as 5C4 (Fig. 3b).

Of the five subtype-specific substitutions located in antigenic site Ø (Supplementary Fig. 5), only position 201 is a major part of the 5C4 epitope. The angle of approach of 5C4 situates Lys201 near the center of the antigen–antibody interface, leading Lys201 to contribute more buried surface area and more potential solvation energy to the binding interaction than any other RSV F residue, as determined using the PISA server[22]. Lys201 is contacted by multiple 5C4 light chain CDRs, and substitution with the Asp found in subtype B viruses would result in a loss of both positive charge and side-chain length at this position, which is predicted to disrupt several hydrogen bonds. Based on the above observations, we hypothesized that disruption to this critical position in RSV F would result in the subtype-specific neutralization observed for this antibody.

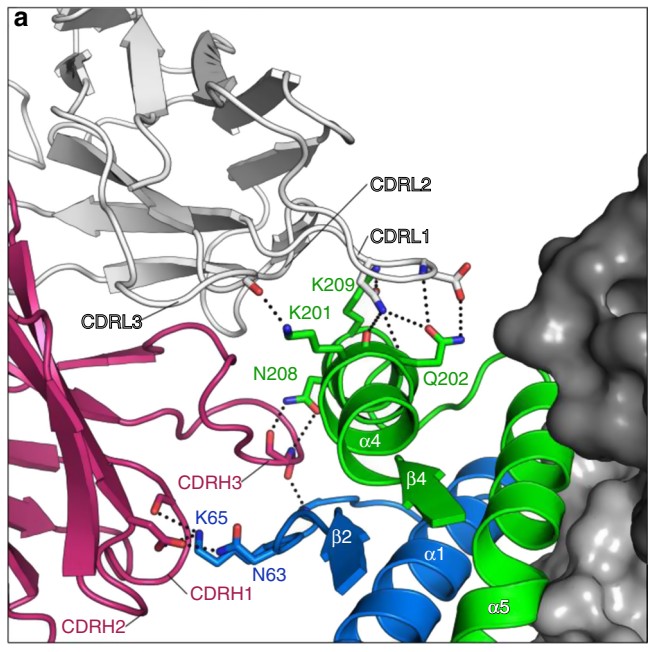

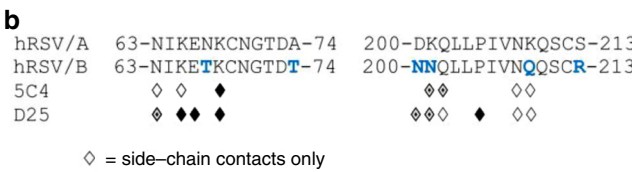

**Fig. 3** 5C4 forms few hydrogen bonds to RSV F main-chain atoms. **a** Two pre-F protomers are shown as dark and light gray surfaces, and the third protomer is shown as ribbons with F1 colored green and F2 colored blue. 5C4 heavy and light chains are shown as ribbons with the heavy and light chains colored pink and white, respectively. Antibody complementarity-determining regions (CDRs) involved in molecular recognition are labeled. Main-chain and side-chain atoms involved in molecular recognition are shown in stick representation with oxygen atoms colored red and nitrogen atoms colored blue. Hydrogen bonds and salt bridges are depicted as black dotted lines. **b** Linear sequence of antigenic site Ø for both RSV subtypes with the F2 β2–α1 loop on the left and the F1 α4 helix on the right. Subtype-specific residues in site Ø are colored blue in the subtype B consensus sequence. RSV F residues that make hydrogen bonds to 5C4 and D25 are denoted with symbols (side-chain hydrogen bonds, main-chain hydrogen bonds, or side-chain and main-chain hydrogen bonds are represented as open, filled, or partially filled diamonds, respectively)

**Antigenic determinants of 5C4 subtype specificity.** To test our hypothesis that 5C4 subtype specificity is primarily due to a single amino acid substitution at RSV F position 201, we quantified 5C4 binding to a panel of single amino acid F variants for each of the five subtype-dependent residues in site Ø, as well as for two strain-dependent residues in this antigenic site. To generate this panel, we grafted single residues from the subtype B strain 18537 to the subtype A strain A2 (referred to as A-chimera constructs) RSV F stabilized in the pre-F conformation by the DS-Cav1 mutations[23]. We then expressed these full-length pre-F variants in Expi293 cells and performed flow cytometry to determine the fractional binding, relative to motavizumab (an affinity matured variant of palivizumab), of 5C4 and D25 IgG to the A-chimera variants. The results indicated that 5C4 could bind each of the A-chimera F proteins with similar efficiency as D25 except for the K201N variant, which was shown to completely ablate 5C4 binding (Fig. 4a). As expected, with its observed breadth against

both RSV subtypes, none of the A-chimera substitutions had substantial impact on D25 binding. To test if grafting any of the subtype A-specific or strain-dependent residues to subtype B-derived F could restore 5C4 binding, we generated an inverse panel to the above, swapping single residues from A2 to 18537 (referred to as B-chimera constructs). Grafting subtype A residue Lys201 to 18537 (variant N201K) restored 5C4 binding activity to near wild-type A2-binding levels (Fig. 4b), further demonstrating the importance of this position to 5C4 binding.

We next sought to determine the effects of a more disruptive or a more subtle substitution at position 201. As expected, the alanine substitution at position 201 ablated 5C4 binding to subtype A-derived pre-F and did not improve 5C4 binding to subtype B-derived pre-F. In contrast, substitution of RSV F position 201 with arginine allowed the retention of 5C4 binding to subtype A-derived pre-F (Fig. 4c) and endowed 5C4 with reactivity toward subtype B-derived pre-F (Fig. 4d). These data demonstrate that a long, positively charged side chain at position 201 is critical for RSV F recognition by 5C4.

**5C4 neutralization of mutant RSV subtype A and B viruses.** To confirm that the results from our binding experiments translated to antibody-mediated virus neutralization, we performed micro-neutralization experiments with recombinant viruses containing subtype-specific substitutions at position 201 for both RSV subtypes. Among subtype B strains, 18537 is unusual in that it has an arginine at position 202 within antigenic site Ø. To account for this strain-dependent variation, we also included a R202Q reversion for these experiments so that our results would be broadly applicable to subtype B strains (Supplementary Fig. 5). We determined the neutralization activity of 5C4 against RSV subtype A and B wild-type and position 201 mutant viruses (Fig. 4e, f). D25 and AM22 were used as controls because they bind the same antigenic site as 5C4 (Supplementary Fig. 6). 5C4 neutralized wild-type A2 virus with high potency (EC$_{50}$ = 0.004 µg/mL), but the K201N substitution in this viral background diminished 5C4 activity 100-fold (EC$_{50}$ = 0.41 µg/mL). Similarly, 5C4 poorly neutralized the wild-type 18537 virus with an EC$_{50}$ of 9.149 µg/mL, yet was able to neutralize the N201K/R202Q 18537 virus with an EC$_{50}$ of 0.022 µg/mL. Taken together, these results confirmed that the effects of the subtype substitutions at position 201 on antibody binding translated to virus neutralization.

**Discussion**

Here we present comparative structural, biochemical, and virological studies of 5C4 and D25, which reveal a molecular basis for subtype-dependent RSV neutralization. Overall features of antibody recognition of site Ø are conserved between mAbs D25 and 5C4, suggesting that there may be a common immunological solution for binding to this neutralization-sensitive epitope[19]. Specifically, the antibody combining sites each form a pincer that grasps the α4 helix of RSV F, and the CDRH3s insert between the α4 helix and the β2–α1 loop. The observation that multiple antibodies bind to antigenic site Ø at different angles and rotations[13] suggests that site Ø is an antigenic supersite[24] of vulnerability on RSV F.

The differences in molecular recognition that account for the observed subtype specificity of 5C4 are only revealed at atomic resolution. D25 is a subtype cross-reactive antibody that forms hydrogen bonds with more than twice as many main-chain atoms in α4 as 5C4, and is especially reliant on the CDRH3. Since main-chain hydrogen bond contacts are generally invariant to sequence changes, D25 may be better able to accommodate sequence variation between subtype A and B strains. In addition, the positioning of the subtype-dependent residue 201 near the center of

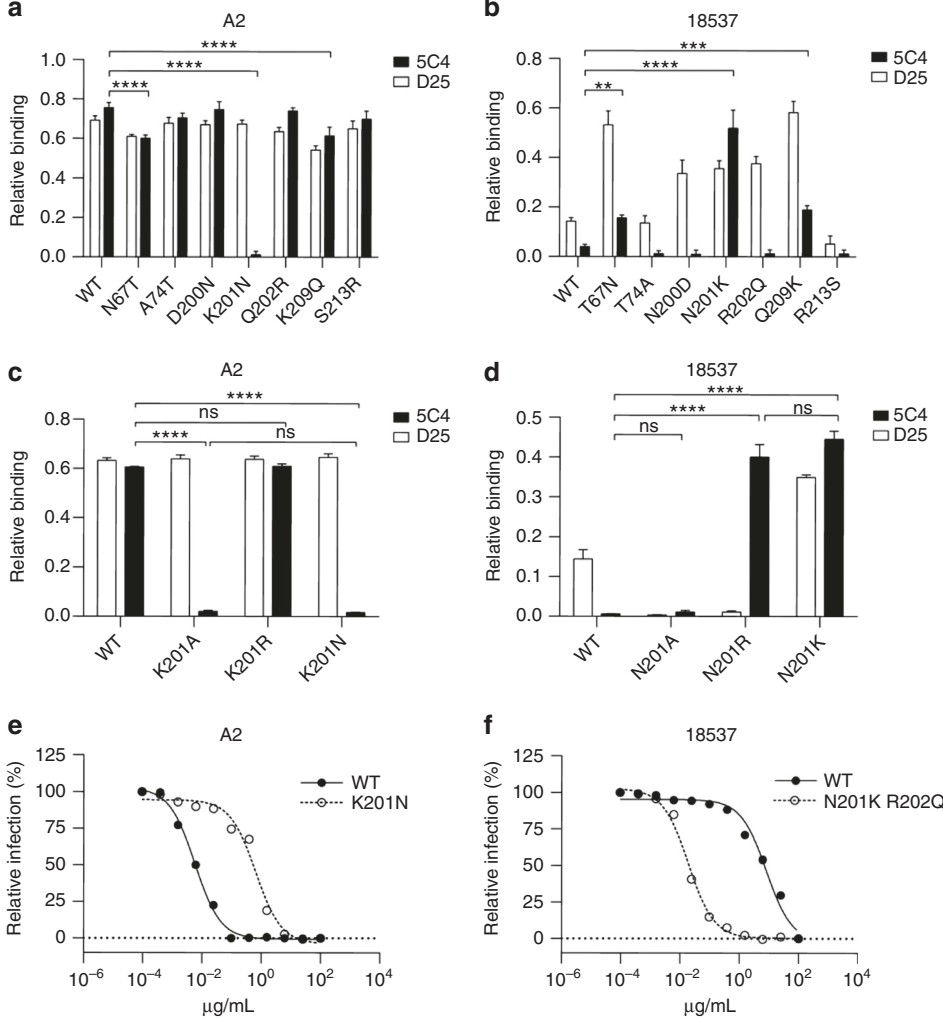

**Fig. 4** Arg or Lys at position 201 is critical for RSV F recognition and neutralization by 5C4. **a–d** Fractional binding of mAbs 5C4 (black bars) and D25 (white bars) to DS-Cav1–stabilized pre-F variants relative to motavizumab. **a** Binding to A-chimera constructs (RSV A2 pre-F variants). **b** Binding to B-chimera constructs (RSV B18537 pre-F variants). **c** Binding to RSV A2 pre-F position 201 variants. **d** Binding to RSV B strain 18537 pre-F position 201 variants. Bars represent the standard relative fractional binding of three replicates with error bars indicating standard deviation. *P* values calculated using two-way ANOVA are shown with symbols, where ns is $P > 0.05$, **$P \leq 0.01$, ***$P \leq 0.001$ and ****$P \leq 0.0001$. **e, f** Neutralization activity of 5C4 against subtype A strain A2 and subtype B strain 18537 with and without subtype-specific substitutions at position 201. **e** The $EC_{50}$ of 5C4 against A2 wild-type and K201N-mutated virus is $0.004 \pm 0.0008$ μg/mL (Mean ±SD) and $0.41 \pm 0.11$ μg/mL, respectively, from three independent experiments. **f** The $EC_{50}$ of 5C4 against RSV B strain 18537 wild-type and N201K-mutated virus is $9.149 \pm 0.056$ μg/mL and $0.022 \pm 0.003$ μg/mL, respectively. Data are representative from three independent experiments

the 5C4–RSV F interface is likely a major factor in the observed subtype specificity of 5C4.

Based on the observation that 5C4 neutralization is subtype-dependent, and the structural insight that 5C4 contacts the subtype-variable position 201 using residues from multiple CDRs, we hypothesized that replacing the subtype A residue Lys201 with the subtype B residue Asn201 would disrupt 5C4-mediated virus neutralization. Binding and neutralization experiments supported our hypothesis, and we further demonstrated that grafting the subtype A Lys201 residue to this position in subtype B improved 5C4 binding and neutralization for subtype B virus. A somewhat similar side-chain sensitivity has been observed previously for hepatitis E virus (HEV)-specific antibody 8C11, which contacts a positively charged side chain, Arg512, on the HEV capsid protein E2 using residues from multiple CDRs[25]. Alteration of this position was demonstrated to abolish the binding interaction with 8C11. However, as Arg512 in E2 is conserved among HEV genotypes, determinants of HEV genotype-specificity were found to be more subtle than in the current study.

An important implication of our results involves the elicitation of antibody responses in RSV seronegative neonates and infants —populations that are most vulnerable to this virus. Our recent studies profiling the human antibody response to RSV F in healthy adults, who have likely been infected by RSV multiple times and perhaps with both RSV A and B viruses, have shown that the majority of mAbs generated against RSV F are subtype cross-reactive, although rare mAbs have been identified that appear to be subtype specific or subtype preferring[14]. This result is supported in the present work, as D25 was isolated from an adult who was likely infected with RSV strains of both subtypes, and this mAb has been shown to have a balanced neutralizing response toward all tested RSV strains. However, although RSV has a single serotype, it has been shown that, in infants, seroconversion of the RSV neutralizing response is skewed toward the infecting viral subtype[26]. This suggests that RSV seronegative infants infected with RSV strains of one subtype may have a reduced ability to mount effective neutralizing responses against RSV strains of the other subtype. As 5C4 was elicited against a

single RSV subtype, some of the subtype-dependent features of this mAb could mimic antibodies elicited during a primary infection in RSV seronegative infants. We posit that, moving forward, vaccine and therapeutic antibody candidates targeted for infants should be carefully designed to provide optimal protection against both RSV subtypes, as small differences in subtype potency of the elicited antibodies could cause neutralization titers to fall below protective thresholds. Vaccine strategies may include co-vaccination or serial vaccination with antigens specific to both subtypes, or vaccination with a chimeric RSV F antigen.

## Methods

**Expression and purification of the 5C4 Fab**. Plasmids encoding 5C4 Fab heavy and light chains[13] were transiently co-transfected into a suspension of Expi293 cells (Invitrogen, catalog# A14527). After incubation at 37 °C for 6 days with shaking, the cell supernatants were passed over a column of CaptureSelect LC-kappa (murine) affinity matrix (Life Technologies). The resin was washed with phosphate-buffered saline (PBS), and 5C4 Fab was eluted with 100 mM glycine pH 3.0 into a solution consisting of 1/10th the elution volume of 1 M Tris–HCl pH 8.0.

**Expression and purification of the 5C4–RSV F complex**. Plasmids encoding 5C4 Fab heavy and light chains were transiently co-transfected into a suspension of Expi293 (Invitrogen) cells with a previously constructed plasmid encoding RSV F (+) FdTHS based on strain A2[13]. After incubation at 37 °C for 6 days with shaking, the cell supernatants were concentrated and buffer exchanged into Ni-NTA binding buffer (20 mM Tris pH 8.0, 20 mM imidazole, 300 mM NaCl) using tangential flow filtration. The 5C4–RSV F complex was purified using manufacturer protocols over a column of Ni-NTA Superflow resin (Qiagen). The Ni-NTA elution was then concentrated and further purified over a column of Strep-Tactin resin (IBA) using the manufacturer's protocol. Following elution off the Strep-Tactin column, C-terminal purification tags on RSV F were removed by overnight digestion at 4 °C with restriction-grade thrombin (EMD Millipore). The complex was then concentrated and further purified to remove excess 5C4 Fab and purification tags by gel filtration using a Superose 6 column (GE Healthcare Biosciences) with a running buffer of 2 mM Tris pH 8.0, 200 mM NaCl and 0.02% NaN$_3$. Fractions corresponding to the 315 kDa complex (Supplementary Fig. 2) were pooled and concentrated for crystallization.

**Crystallization and X-ray data collection**. Crystals of 5C4 Fab were produced by the hanging-drop vapor-diffusion method by mixing 1 μL of 5C4 Fab (8.5 mg/mL) with 1 μL of reservoir solution containing 0.1 M sodium acetate trihydrate pH 4.6 and 20% PEG 3350. Crystals of 5C4 Fab were transferred to a cryoprotectant solution (0.1 M sodium acetate trihydrate pH 4.6, 20% PEG 3350, 15% 2R,3R-butanediol) and were flash frozen in liquid nitrogen. Crystals of the 5C4–RSV F complex were produced by the vapor-diffusion method along with additive screening and microseed matrix screening (MMS)[27]. Initial crystals of the 5C4–RSV F complex were produced by mixing 0.5 μL of 5C4–RSV F complex (4.9 mg/mL) with 1 μL of reservoir solution containing 0.01 M zinc sulfate heptahydrate, 0.1 M MES hydrate pH 6.5, 22.5% PEG MME 550 and 0.5% polyvinylpyrrolidone K15. These initial crystals were then smashed with a microtool and transferred into a 1.5 mL microtube containing a seed bead and stabilizing solution (25% PEG MME 550, 0.01 M zinc sulfate, 0.1 M MES pH 6.5). The tube was vortexed for two min, kept cold, and placed on ice. An Index screen (Hampton Research) was set up using the NT8 robot (Formulatrix), with each drop consisting of 150 nL protein (4.9 mg/mL), 50 nL seed stock (above) and 100 nL of reservoir solution. The condition that generated the single crystal that diffracted to 3.4 Å consisted of 0.2 M ammonium acetate, 0.1 M Bis-Tris pH 6.5 and 25% PEG 3350. Crystals were transferred to a cryoprotectant solution (0.2 M ammonium acetate, 0.1 M Bis-Tris pH 6.5, 25% PEG 3350, 15% 2R,3R-butanediol) and were then flash frozen in liquid nitrogen. X-ray diffraction data for the 5C4–RSV F complex were collected to 3.4 Å at the CHESS beamline A1 (Cornell High Energy Synchrotron Source). X-ray diffraction data for the 5C4 Fab were collected to 1.5 Å at the NSLS beamline X6A (National Synchrotron Light Source).

**Structure determination, model building and refinement**. Diffraction data were indexed and integrated in iMOSFLM[28] and scaled and merged with AIMLESS[29]. A molecular replacement solution for the 5C4 Fab was obtained by PHASER[30] using a murine antibody heavy chain from PDB ID 3T3P[31] and light chain from PDB ID 4GAG[32]. A molecular replacement solution for the 5C4–RSV F complex was obtained by PHASER using pre-F from PDB ID 5C69[20] and the refined 5C4 Fab as search models. The structures were built manually in COOT[33] and refined using PHENIX[34]. Data collection and refinement statistics are presented in Supplementary Table 1.

**Expression and purification of 6xHis-tagged RSV F proteins**. Freestyle 293-F cells (Invitrogen, catalog # R79007) were transfected with previously described pαH-backbone plasmid encoding stabilized StrepTagII and 6xHis-tagged RSV pre-F (DS-Cav1) in the A2 or 9320 RSV F background[13, 23]. Proteins were expressed in the presence of kifunensine (5 μM) and purified from supernatants using Strep-Tactin resin (IBA). Parental pαH-backbone plasmid was obtained from the Vaccine Research Center (VRC) at the National Institutes of Health (NIH).

**Surface plasmon resonance experiments**. 6xHis-tagged RSV pre-F (DS-Cav1) proteins in the A2 or 9320 background were captured on a NTA sensor chip to ~150 response units for each cycle using a Biacore X100 (GE Healthcare). After three injections of running buffer, 3-fold dilutions of increasing concentrations of Fabs were injected over both the ligand-bound and reference flow cells. (5C4:pre-F A2, 0.046–300 nM; 5C4:pre-F 9320, 0.460–3000 nM; D25:pre-F A2, 0.0152–100 nM; D25:pre-F 9320, 0.0152–100 nM). The chip was regenerated twice after each cycle using 350 mM EDTA followed by 0.5 mM NiCl$_2$. Data were double-reference subtracted and fit to a 1:1 binding model using Biacore analysis software.

**Cloning of full-length RSV F expression plasmids**. DNA encoding full-length RSV F (TM, residues 1–574) from strains A2 and 18537 were codon optimized for human expression and synthesized with the DS-Cav1 prefusion-stabilizing mutations (Supplementary Note 1)[13]. These TM constructs were subcloned into pVRC8400 expression plasmid[35]. Site-directed mutagenesis was used to generate variants of the full-length RSV F A2 and 18537 expression plasmids, including A- and B-chimera constructs, as well as constructs encoding for altered amino acids at position 201.

**Cell-surface binding assays**. Plasmids expressing stabilized RSV F TM construct variants were transiently transfected into Expi293 cells. Forty-eight hours post-transfection, 6 million cells were harvested, trifurcated evenly, and immediately stained with either Alexa Fluor 488-conjugated D25 IgG, unconjugated 5C4, or unconjugated motavizumab IgG for 1 h at 4 °C and then washed. 5C4-stained cells were further stained with FITC-conjugated goat anti-mouse secondary antibody (Virus Research Core, NIAID, NIH) and motavizumab-stained cells were further stained with Alexa Fluor 488-conjugated goat anti-human secondary antibody (Invitrogen) for 30 min at 4 °C and then washed. After staining was complete, cells were fixed with 0.5 % paraformaldehyde and then analyzed on a 5-laser LSRFortessa™ flow cytometer (Becton Dickinson). Data analysis was performed using FLOWJO version 7.6.5 (Treestar Inc.).

**Generation and recovery of recombinant viruses**. A bacterial artificial chromosome (BAC) system[36], pSynkRSV-line19F, was used to generate recombinant RSV antigenomes by transferring F sequence corresponding to wild-type RSV lab-strain A2 (accession# KT992094), pSynkRSV-A2F; clinical isolates 18537 (accession# D00334), pSynkRSV-18537F and TX11-56 (G: accession# JQ680989; F: accession# JQ736679), pSynkRSV-TX11-56 F. Mutant RSV antigenomes were also generated for RSV A2 containing the K201N substitution (pSynkRSV-A2F-K201N) and RSV 18537 containing the N201K and R202Q substitutions (pSynkRSV-18537F-N201K-R202Q). These BACs, as well as sequence-optimized helper plasmids encoding the RSV N, P, M2-1 and L genes[35], were co-transfected into BSR-T7/5 cells[37]. Transfected cells were cultured at 37 °C and passaged every 3–4 days until cells began to form syncytia. When single large syncytia were visible, infectious virus was recovered and used to infect healthy HEp-2 cells (ATCC, catalog #CCL-23). This process was iterated for 4 rounds of selection before purified virus was used to generate master and working stocks.

**RSV neutralization assays**. Antibody neutralization was measured using a fluorescent plate-reader assay[23]. Briefly, HEp-2 cells were infected with mKate-RSV strains A2, 18537, A2–K201N, or 18537–N201K in the presence of 3- or 4-fold serial antibody dilutions. Infection was monitored as a function of mKate expression at 22–24 h post-infection in a Spectramax fluorescence plate reader (Molecular Devices Inc.) with an excitation wavelength of 588 nm and an emission wavelength of 635 nm. Data were analyzed by curve fitting and non-linear regression in PRISM (GraphPad Software Inc.) to determine EC$_{50}$ as well as percent neutralization at a given antibody concentration.

**Data availability**. Coordinates and structure factors for the 5C4–RSV F complex and the 5C4 Fab have been deposited in the Protein Data Bank under accession codes 5W23 and 5W24, respectively. The authors declare that all other data supporting the findings of this study are available within the article and its Supplementary Information files, or are available from the authors upon request.

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

## Acknowledgements

We would like to thank members of the Graham and McLellan labs for comments on the manuscript, Bryan Neumann for identifying the initial 5C4 Fab crystallization condition, Emilie Shipman for assistance with protein expression and purification, and members of the A1 beamline staff at CHESS and members of the X6A beamline staff at NSLS for assistance with X-ray data collection. CHESS is supported by NSF award DMR-1332208, and the MacCHESS resource is supported by NIGMS award GM-103485. Use of the NSLS is supported by the U.S. Department of Energy, Office of Science, Office of Basic Energy Sciences under Contract No. DE-AC02-98CH10886. Support for this work was provided by grants from the National Natural Science Foundation of China 81361120408 and 81401668 (N.–S.X.), National Institutes of Health P20GM113132 (J.S.M.) and 5T32AI007519 (M.B.B.), and intramural funding from the National Institute of Allergy and Infectious Diseases to support work at the NIAID Vaccine Research Center (B.S.G.).

## Author contributions

D.T. cloned full-length RSV F variants, performed the cell-surface binding analysis, rescued viruses with substitutions at position 201, and completed the neutralization studies. M.B.B. expressed and purified 5C4 and the 5C4–RSV F complex, performed Biacore experiments, and built and refined the 5C4–RSV F structure. S.M.M. generated the clones and rescued RSV subtype A and B wild-type and mutant viruses, and performed neutralization assays and mutagenesis. A.L.H. and M.L.M. provided the plasmids for rescue of the L19 virus. M.C., K.M., A.K., M.K. and K.W.G. produced reagents and participated in analysis of binding and neutralization assays. N.M.T. built and refined the 5C4 Fab structure. M.Z., Z.Z. and N.–S.X. provided reagents and contributed to study design and analysis. J.S.M and B.S.G conceived and designed the study, and along with D. T. and M.B.B., analyzed the data and wrote the manuscript. All authors discussed and commented on the manuscript.

## Additional information

**Competing interests:** M.C., M.Z., Z.Z., N.–S.X., J.S.M. and B.S.G. are listed as co-inventors on a patent application describing the discovery and utility of 5C4 (PCT/CN2014/073505). The remaining authors declare no competing financial interests.

