## [Peer Review File · Nature Communications]

Reviewers' comments:

Reviewer #1 (Remarks to the Author):

In this manuscript, Tian et al. characterized a RSV F subtype A-preferred antibody 5C4 and determined its complex structure with the RSV F protein. The crystal structure shows that 5C4 shares the similar epitope site \emptyset with previously reported antibody D25, but 5C4 forms fewer hydrogen bonds to the main chain atoms. By structural analysis and mutagenesis studies, the authors identified a key residue 201 that largely governs the subtype specificity of 5C4. Most of previously reported RSV F antibodies bind and neutralize both RSV F subtype A and B, hence the structure of subtype A-specific antibody 5C4 is particularly valuable for understanding the structural basis of viral subtype-dependent antibody neutralization, and would provide guidance for future therapeutic antibody candidates selection and RSV vaccine design to elicit less subtype-specific antibodies. The manuscript is well written and very clear. Some specific comments are as follows;

Main point:

The authors proposed a mechanism of subtype-specific recognition and neutralization of 5C4, as shown in figure 3B, "D25, which neutralizes A and B subtypes equally well, forms hydrogen bonds with more than twice as many main-chain atoms in RSV F as 5C4. Since main-chain hydrogen bond contacts are generally invariant to sequence changes, 5C4 appears less able to compensate for the loss of side-chain binding contacts when interacting with RSV F subtype B" (line 140-143).

(1) Although the above logic sounds reasonable, the authors did not consider that the contribution of each hydrogen bond to the antibody-antigen interactions varies a lot. A loss of one hydrogen bond or salt-bridge interaction at a critical position would markedly impair the antibody binding to the antigen. If D25 is more tolerate to the sequence variations because of making more hydrogen bond with the main-chains, then the side-chain change of a single residue should not affect the D25 binding to the antigen dramatically, which appear not to be always true. For example, the N201A abolished the D25 binding to 18537 (Figure 4D), whereas T67N significantly enhanced the D25 binding to 18537 (Figure 4B).

(2) As the authors mentioned, "only position 201 is a major part of the 5C4 epitope and contributes more buried surface area and more potential solvation energy to the binding interaction" (line 145), "Lys201 is contacted by multiple 5C4 light chain CDRs"(line 147), therefore the K201 here seems to be a very critical residue for 5C4 binding. From the structure comparison, since there is a ~ 45 degree angle between 5C4 and D25, it also looks like 5C4 is more closer to the $\beta 2$ than D25, and the residue 201 is more closer to the center and more buried in 5C4 interface than D25 interface. Therefore, the subtype-specific residue 201 may be more important for 5C4 binding than D25 binding. This might be a more plausible explanation to the subtype specificity of 5C4.

(3) Generation of 5C4 resistant mutants and looking for D25 activity and vice versa (D25 resistant mutants and 5C4 activity) could also help refining the role of individual contact residues (main chain vs. side-chains) contribution in explaining the specificity

Reviewer #2 (Remarks to the Author):

This manuscript describes the crystal structures for a subtype-dependent broadly neutralizing antibody (5C4) against the RSV fusion protein, and that antibody in complex with the fusion protein trimer. The structure of the complex is compared to that previously determined with the more broadly neutralizing antibody D25, and the structures give us some idea of why one antibody is superior to the other in its breadth. Mutagenesis and neutralization assays are included to show that F residue 201 is the determinant of specificity in 5C4. The paper is clear and well written and will be of high interest for those developing anti-RSV vaccines, and for anyone interested in antibody-virus interactions. Specific comments:

In the supplemental table of crystallographic data, the Rmerge in the high-resolution shell for the complex is too low- please check. In this table, please include Rmeas, Rpim, and the clashscore. Also, it is not clear why there are 48 'ligand' atoms for the complex structure, when the validation report just mentions 8 zinc atoms. For both the unliganded and complex structure I would recommend changing the 'ligand' designator in this table to 'solvent' or something similar, as neither zinc or butanediol are really ligands.

I believe that the fusion protein is glycosylated, but there are no mentions of carbohydrates in the paper. Are these present at all? If so, is any electron density visible? Or is the protein treated with any sort of deglycosylating enzyme before crystallization?

Reviewer #3 (Remarks to the Author):

The authors, in groundbreaking work, have previously identified a monoclonal antibody (mAb) binding site, site ϕ , that is unique to the pre-fusion form of the respiratory syncytial virus (RSV) F protein and showed that mAbs specific to this site are the most potent RSV neutralizing antibodies. There have been, until just recently, two mAbs specific to site ϕ described in the literature, mAbs 5C4 and D25, and the structure of D25 bound to the pre-fusion F has been previously solved and published.

In this manuscript, the authors have solved the structure of 5C4 bound to the pre-fusion F protein comparing it to the D25-Pre-F complex. They have demonstrated in structural analysis that, while the sites bound by the two antibodies are overlapping, the binding of the two antibodies is somewhat different. What is significant here is that they show that 5C4, in contrast to D25, discriminates between the F protein of RSV subtype A and RSV subtype B. They show that 5C4 poorly neutralizes subtype B viruses while potently neutralizing subtype A viruses. D25 neutralizes both subtypes. Furthermore their detailed structural analyses of the contact points between the pre-fusion F protein and the two antibodies potentially account for this difference in biological activity. Their analyses suggested that amino acid at position 201 in F protein is key to the differences in binding and neutralization. They support their hypothesis with mutational analysis of site ϕ .

The authors' conclusions are well supported by the data presented. What is most significant here is that there are differences in mAb 5C4 neutralization of RSV subtype A and B with a reasonable structural basis for that difference. The RSV F protein has been the primary target of vaccine development for RSV because the dogma in the field has been that the F protein is antigenically conserved between RSV A and B and that immune responses to one

will protect from infection by the other. The data in this manuscript demonstrate that the situation may not be as clear cut as previously thought and that more attention should be paid to immunological differences between subtype A and B F proteins.

Points to consider:

1. Have any 5C4-like mAb derived from humans been identified? One of the authors is an author on an exhaustive characterization of hundreds of anti-F protein mAb derived from human B cells. Do any of these antibodies have the ability to discriminate between subtype A and subtype B? In other words, is a 5C4 type antibody typical only of murine immune responses? This is an important point if these results impact human vaccine development as the authors argue.
2. The authors should compare the K_d of binding of the two antibodies to RSV A and RSV B F proteins.
3. Figure 4: how do the authors account for the low binding of D25 to subtype B wild type compared to subtype A wild type? The authors indicate that the results are normalized to motavizumab binding. Does motavizumab bind equally to subtype A and B Fs?
4. Supplemental Fig 2 should be combined with Fig 2B since it more clearly illustrates the different angles of the binding.
5. Lines 92-93 need revision.

Reviewer #1 (Remarks to the Author):

In this manuscript, Tian et al. characterized a RSV F subtype A-preferred antibody 5C4 and determined its complex structure with the RSV F protein. The crystal structure shows that 5C4 shares the similar epitope site \emptyset with previously reported antibody D25, but 5C4 forms fewer hydrogen bonds to the main chain atoms. By structural analysis and mutagenesis studies, the authors identified a key residue 201 that largely governs the subtype specificity of 5C4. Most of previously reported RSV F antibodies bind and neutralize both RSV F subtype A and B, hence the structure of subtype A-specific antibody 5C4 is particularly valuable for understanding the structural basis of viral subtype-dependent antibody neutralization, and would provide guidance for future therapeutic antibody candidates selection and RSV vaccine design to elicit less subtype-specific antibodies. The manuscript is well written and very clear. Some specific comments are as follows;

Main point: The authors proposed a mechanism of subtype-specific recognition and neutralization of 5C4, as shown in figure 3B, “D25, which neutralizes A and B subtypes equally well, forms hydrogen bonds with more than twice as many main-chain atoms in RSV F as 5C4. Since main-chain hydrogen bond contacts are generally invariant to sequence changes, 5C4 appears less able to compensate for the loss of side-chain binding contacts when interacting with RSV F subtype B” (line 140-143).

(1) Although the above logic sounds reasonable, the authors did not consider that the contribution of each hydrogen bond to the antibody-antigen interactions varies a lot. A loss of one hydrogen bond or salt-bridge interaction at a critical position would markedly impair the antibody binding to the antigen. If D25 is more tolerate to the sequence variations because of making more hydrogen bond with the main-chains, then the side-chain change of a single residue should not affect the D25 binding to the antigen dramatically, which appear not to be always true. For example, the N201A abolished the D25 binding to 18537 (Figure 4D), whereas T67N significantly enhanced the D25 binding to 18537 (Figure 4B).

(2) As the authors mentioned, “only position 201 is a major part of the 5C4 epitope and contributes more buried surface area and more potential solvation energy to the binding interaction” (line 145), “Lys201 is contacted by multiple 5C4 light chain CDRs” (line 147), therefore the K201 here seems to be a very critical residue for 5C4 binding. From the structure comparison, since there is a ~ 45 degree angle between 5C4 and D25, it also looks like 5C4 is more closer to the $\beta 2$ than D25, and the residue 201 is more closer to the center and more buried in 5C4 interface than D25 interface. Therefore, the subtype-specific residue 201 may be more important for 5C4 binding than D25 binding. This might be a more plausible explanation to the subtype specificity of 5C4.

RESPONSE: We thank the reviewer for these observations and suggestions, and agree that we did not properly consider the degree to which each hydrogen bond or salt bridge contributes to the interaction. The prominent nature of Lys201 in the 5C4 epitope may be a more plausible explanation for the subtype specificity of 5C4, and we have therefore made several modifications to the Abstract, Introduction, Results and Discussion:

1) A sentence in the Abstract previously stating the 5C4 makes fewer hydrogen bonds to main chain atoms has been altered to “The crystal structure of 5C4 bound to the RSV fusion (F)

protein revealed that the overall binding mode of 5C4 is similar to that of D25, but their angles of approach are substantially different.”

2) Similar text has been removed from the last paragraph of the introduction.

3) We removed the sentence from the Results section referenced above (original lines 140-143).

4) We modified the sentence from the Results section (original line 145) as follows: “Of the five subtype-specific mutations located in antigenic site Ø (**Supplementary Fig. 4**), only position 201 is a major part of the 5C4 epitope. The angle of approach of 5C4 situates Lys201 near the center of the antigen-antibody interface, leading Lys201 to contribute more buried surface area and more potential solvation energy to the binding interaction than any other RSV F residue, as determined using the PISA server.”

5) We also modified the paragraph in the Discussion section describing the subtype-specificity to read as follows: “The differences in molecular recognition that account for the observed subtype-specificity of 5C4 are only revealed at atomic resolution. D25 is a subtype cross-reactive antibody that forms hydrogen bonds with more than twice as many main-chain atoms in $\alpha 4$ as 5C4, and is especially reliant on the CDRH3. Since main-chain hydrogen bond contacts are generally invariant to sequence changes, D25 may be better able to accommodate sequence variation between subtype A and B strains. Additionally, the positioning of the subtype-dependent residue 201 near the center of the 5C4–RSV F interface is likely a major factor in the observed subtype-specificity of 5C4.”

(3) Generation of 5C4 resistant mutants and looking for D25 activity and vice versa (D25 resistant mutants and 5C4 activity) could also help refining the role of individual contact residues (main chain vs. side-chains) contribution in explaining the specificity

RESPONSE: Thank you for this suggestion, but we have been unable to generate escape mutations in vitro within site Ø despite repeated attempts. We could potentially do a much more extensive mutational analysis, but in the absence of escape mutations we do not think this approach is likely to provide additional insight.

Reviewer #2 (Remarks to the Author):

This manuscript describes the crystal structures for a subtype-dependent broadly neutralizing antibody (5C4) against the RSV fusion protein, and that antibody in complex with the fusion protein trimer. The structure of the complex is compared to that previously determined with the more broadly neutralizing antibody D25, and the structures give us some idea of why one antibody is superior to the other in its breadth. Mutagenesis and neutralization assays are included to show that F residue 201 is the determinant of specificity in 5C4. The paper is clear and well written and will be of high interest for those developing anti-RSV vaccines, and for anyone interested in antibody-virus interactions.

Specific comments:

In the supplemental table of crystallographic data, the Rmerge in the high-resolution shell for the complex is too low- please check. In this table, please include Rmeas, Rpim, and the clashscore. Also, it is not clear why there are 48 ‘ligand’ atoms for the complex structure, when the validation report just mentions 8 zinc atoms. For both the unliganded and complex structure I would recommend changing

the 'ligand' designator in this table to 'solvent' or something similar, as neither zinc or butanediol are really ligands.

RESPONSE: We thank the reviewer for these detailed observations and suggestions. The table of crystallographic data has been updated to include R_{meas} , R_{pim} and the clashscore, and the zinc and butanediol molecules are now counted as "solvent". Additionally, the table has been corrected to reflect the actual R_{merge} value for the inner shell of the complex (1.596) and to reflect the actual number of zinc atoms in the complex (8).

I believe that the fusion protein is glycosylated, but there are no mentions of carbohydrates in the paper. Are these present at all? If so, is any electron density visible? Or is the protein treated with any sort of deglycosylating enzyme before crystallization?

RESPONSE: There are three *N*-linked glycans in the ectodomain of the cleaved F glycoprotein at positions 27, 70, and 500. The F protein used in our crystallographic studies was produced in HEK293 cells with presumably complex glycans and crystallized without endoglycosidase treatment. There is some unmodeled density observed at 3 of the 9 potential glycosylated positions in the current structure, (N27 in chain B, N70 in chain C and N500 in chain A), however, the density at these positions is not sufficient to warrant building the core GlcNAc.

We have included the following sentence at the end of the 2nd paragraph of the Results section to address this query. "Although the F glycoprotein was fully glycosylated, the electron density at each of the *N*-linked glycosylation sites was insufficient to accurately model the glycans."

Reviewer #3 (Remarks to the Author):

The authors, in groundbreaking work, have previously identified a monoclonal antibody (mAb) binding site, site \emptyset , that is unique to the pre-fusion form of the respiratory syncytial virus (RSV) F protein and showed that mAbs specific to this site are the most potent RSV neutralizing antibodies. There have been, until just recently, two mAbs specific to site \emptyset described in the literature, mAbs 5C4 and D25, and the structure of D25 bound to the pre-fusion F has been previously solved and published.

In this manuscript, the authors have solved the structure of 5C4 bound to the pre-fusion F protein comparing it to the D25-Pre-F complex. They have demonstrated in structural analysis that, while the sites bound by the two antibodies are overlapping, the binding of the two antibodies is somewhat different. What is significant here is that they show that 5C4, in contrast to D25, discriminates between the F protein of RSV subtype A and RSV subtype B. They show that 5C4 poorly neutralizes subtype B viruses while potentially neutralizing subtype A viruses. D25 neutralizes both subtypes. Furthermore their detailed structural analyses of the contact points between the pre-fusion F protein and the two antibodies potentially account for this difference in biological activity. Their analyses suggested that amino acid at position 201 in F protein is key to the differences in binding and neutralization. They support their hypothesis with mutational analysis of site \emptyset .

The authors' conclusions are well supported by the data presented. What is most significant here is that there are differences in mAb 5C4 neutralization of RSV subtype A and B with a reasonable structural basis for that difference. The RSV F protein has been the primary target of vaccine development for RSV because the dogma in the field has been that the F protein is antigenically conserved between RSV A and B and that immune responses to one will protect from infection by the other. The data in this

manuscript demonstrate that the situation may not be as clear cut as previously thought and that more attention should be paid to immunological differences between subtype A and B F proteins.

Points to consider:

1. Have any 5C4-like mAb derived from humans been identified? One of the authors is an author on an exhaustive characterization of hundreds of anti-F protein mAb derived from human B cells. Do any of these antibodies have the ability to discriminate between subtype A and subtype B? In other words, is a 5C4 type antibody typical only of murine immune responses? This is an important point if these results impact human vaccine development as the authors argue.

RESPONSE: There are mAbs isolated from adults that we have found that selectively bind and neutralize either subtype A or B fusion glycoproteins. Some of these are site \emptyset specific, but the large majority bind pre-F from subtype A and B equally well. Therefore, it is not an artifact of mice, but it is also not common in adults infected multiple times throughout their lives, likely by both A and B viruses.

We have modified the following sentence in the last paragraph of the Discussion section to address this concern. "Recent work from our labs' profiling of the human antibody response to RSV F in healthy adults, who have likely been infected by RSV multiple times and perhaps with both RSV A and B viruses, has shown that the majority of mAbs generated against RSV F are subtype cross-reactive, although rare mAbs have been identified that appear subtype-specific or subtype-preferring¹⁴."

2. The authors should compare the Kd of binding of the two antibodies to RSV A and RSV B F proteins.

RESPONSE: Thank you for this suggestion. We have now added this data as Supplementary Figure 1 and have modified the following sentence in the 1st paragraph of the Results section. "However, when analyzed against a panel of RSV strains including both subtypes A and B, it was revealed that 5C4 had significantly reduced neutralization potency against various subtype B viruses (Fig. 1a, b) as well as reduced affinity towards pre-F derived from subtype B strain 9320, as measured by surface plasmon-resonance (Supplementary Fig. 1)."

3. Figure 4: how do the authors account for the low binding of D25 to subtype B wild type compared to subtype A wild type? The authors indicate that the results are normalized to motavizumab binding. Does motavizumab bind equally to subtype A and B Fs?

RESPONSE: We are also intrigued by the finding that D25 binding to F from the 18537 B strain relative to motavizumab is weaker than to F from the A2 strain in the flow cytometry-based assay even though neutralization potency against A2 and 18537 strains is similar. We have evaluated this in different formats. As noted above surface plasmon resonance estimate of Kd is now shown in Supplementary Fig. 1 and shows similar binding of D25 to both subtypes. Motavizumab binds both subtypes equally well by Biacore and Octet. Motavizumab binding of F-transduced 293 cells is a little higher in the B subtype. We have repeated the flow cytometry experiments shown in Figure 4 with identical results so we are confident of the relative binding in that assay. We also evaluated AM14 binding in the repeat experiment to ensure that amino

acid modifications were not making F less stable resulting in spontaneous rearrangement into the post-F conformation on the cell surface. AM14 recognizes a quaternary epitope and is the most sensitive test for stability of the pre-F conformation. These experiments demonstrated that the 18537 B strain F is predominantly in the pre-F conformation. Therefore, we do not have a good explanation for this finding, but as we have started identifying additional site \emptyset monoclonal antibodies we do see a range of subtype binding preferences with most binding equally well to both subtypes A and B.

4. Supplemental Fig 2 should be combined with Fig 2B since it more clearly illustrates the different angles of the binding.

RESPONSE: Thank you for this suggestion. Supplementary Figure 2 has now been added to Figure 2b to better demonstrate the different angle of binding.

5. Lines 92-93 need revision.

RESPONSE: We thank the reviewer for pointing out this typo. The error has been corrected.

REVIEWERS' COMMENTS:

Reviewer #1 (Remarks to the Author):

The manuscript is considerably improved and acceptable for publication

Reviewer #2 (Remarks to the Author):

I am satisfied that the authors have addressed the reviewers concerns and recommend publication of the manuscript.

Reviewer #3 (Remarks to the Author):

This manuscript is a revised version of a manuscript previously submitted. The authors have carefully considered the comments of all three reviewers, revising their document appropriately. I am satisfied with their revisions and have no further comments.

REVIEWERS' COMMENTS:

Reviewer #1 (Remarks to the Author):

The manuscript is considerably improved and acceptable for publication

Reviewer #2 (Remarks to the Author):

I am satisfied that the authors have addressed the reviewers concerns and recommend publication of the manuscript.

Reviewer #3 (Remarks to the Author):

This manuscript is a revised version of a manuscript previously submitted. The authors have carefully considered the comments of all three reviewers, revising their document appropriately. I am satisfied with their revisions and have no further comments.

RESPONSE: Thank you for reviewing our revised manuscript. We are glad that you are satisfied by our technical revisions and find the manuscript acceptable for publication. We have addressed the additional editorial comments as noted in the cover letter and expanded the methods to incorporate more information on the methodology, added stereo images for the crystallographic structures as new Supplementary Figure 2, included sequences for the newly generated RSV Fs as Supplementary Data 1, and made additional minor edits to conform to the Nature Communications format.